# Live imaging and biophysical modeling support a button-based mechanism of somatic homolog pairing in *Drosophila*

Myron Barber Child VI[1,2†], Jack R Bateman[3†*], Amir Jahangiri[4], Armando Reimer[5], Nicholas C Lammers[5], Nica Sabouni[1], Diego Villamarin[3], Grace C McKenzie-Smith[3], Justine E Johnson[3], Daniel Jost[4,6*], Hernan G Garcia[1,2,5,7*]

[1]Department of Molecular and Cell Biology, University of California, Berkeley, Berkeley, United States; [2]Department of Physics, University of California, Berkeley, United States; [3]Biology Department, Bowdoin College, Brunswick, United States; [4]Univ Grenoble Alpes CNRS, Grenoble INP, TIMC-IMAG, Grenoble, France; [5]Biophysics Graduate Group, University of California, Berkeley, Berkeley, United States; [6]Université de Lyon, ENS de Lyon, Univ Claude Bernard, CNRS, Laboratory of Biology and Modeling of the Cell, Lyon, France; [7]Institute for Quantitative Biosciences-QB3, University of California, Berkeley, Berkeley, United States

*For correspondence:
jbateman@bowdoin.edu (JRB);
daniel.jost@ens-lyon.fr (DJ);
hggarcia@berkeley.edu (HGG)

†These authors contributed
equally to this work

Competing interests: The
authors declare that no
competing interests exist.

Reviewing editor: Pierre Sens,
Institut Curie, PSL Research
University, CNRS, France

**Abstract** Three-dimensional eukaryotic genome organization provides the structural basis for gene regulation. In *Drosophila melanogaster*, genome folding is characterized by somatic homolog pairing, where homologous chromosomes are intimately paired from end to end; however, how homologs identify one another and pair has remained mysterious. Recently, this process has been proposed to be driven by specifically interacting 'buttons' encoded along chromosomes. Here, we turned this hypothesis into a quantitative biophysical model to demonstrate that a button-based mechanism can lead to chromosome-wide pairing. We tested our model using live-imaging measurements of chromosomal loci tagged with the MS2 and PP7 nascent RNA labeling systems. We show solid agreement between model predictions and experiments in the pairing dynamics of individual homologous loci. Our results strongly support a button-based mechanism of somatic homolog pairing in *Drosophila* and provide a theoretical framework for revealing the molecular identity and regulation of buttons.

## Introduction

Eukaryotic genomes are highly organized within the three-dimensional volume of the nucleus, from the large scale of chromosome territories to the smaller-scale patterned folding of chromosomal segments called topologically associated domains (TADs) and the association of active and inactive chromatin into separate compartments (*Szabo et al., 2019*). Disruption of these organizational structures can have large consequences for gene expression and genome stability (*Lupiáñez et al., 2015*; *Kragesteen et al., 2018*; *Despang et al., 2019*; *Rosin et al., 2019*), emphasizing the importance of fully understanding the mechanisms underlying three-dimensional genome organization.

While many principles of genome organization are common among eukaryotes, differences have been noted between organisms and cell types. For example, in somatic cells in *Drosophila*, an additional layer of nuclear organization exists: homologous chromosomes are closely juxtaposed from end to end, a phenomenon known as somatic homolog pairing (*Joyce et al., 2016*; *Stevens, 1908*). While similar interchromosomal interactions occur transiently in somatic cells of other species and

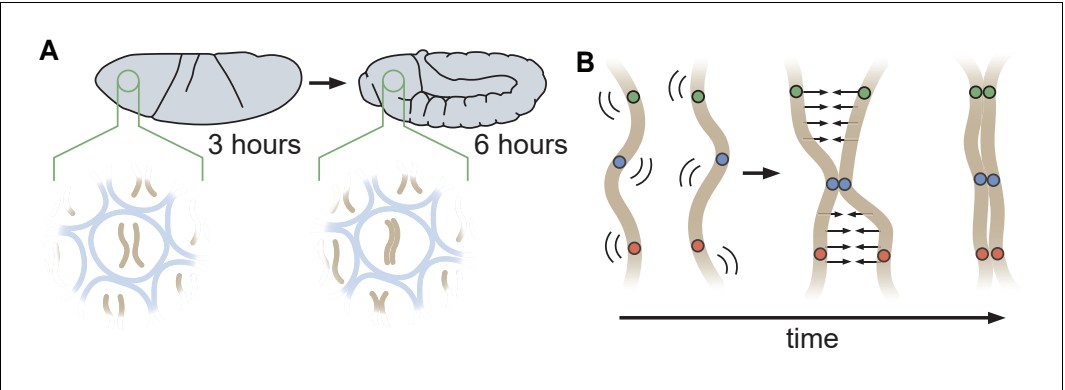

**Figure 1.** Schematic of homologous chromosome pairing in somatic cells in *D. melanogaster*. (A) Over the course of embryonic development, homologous chromosomes pair along their lengths. (B) Button model for homolog pairing in which each chromosome carries a series of sites that have affinity for the same site on its homologous chromosome.

during early meiotic phases of most sexually reproducing eukaryotes, the widespread and stable pairing of homologous chromosomes in somatic cells of *Drosophila* appears to be unique to Dipteran flies (*King et al., 2019*; *Joyce et al., 2016*; *McKee, 2004*). Notably, the close juxtaposition of paired homologs can have a dramatic impact on gene expression through a process known as transvection, whereby regulatory elements on one chromosome influence chromatin and gene expression on a paired chromosome (*Fukaya and Levine, 2017*; *Duncan, 2002*). Although somatic homolog pairing was first described over 100 years ago (*Stevens, 1908*), the molecular mechanisms by which homologous chromosomes identify one another and pair have yet to be described.

During the early stages of *Drosophila* development, maternal and paternal genomes are initially separated and become paired as embryogenesis proceeds. Prior analyses of the initiation of somatic homolog pairing have relied primarily on DNA fluorescent in situ hybridization (DNA-FISH) to label homologous loci in fixed embryos, and have led to a model in which somatic homolog pairing slowly increases with developmental time through independent associations along the lengths of each chromosome arm (*Figure 1A*; *Fung et al., 1998*; *Hiraoka et al., 1993*; *Gemkow et al., 1998*). This model is further supported by recent studies that converged on a 'button' model for pairing, which hypothesizes that pairing is initiated at discrete sites along the length of each chromosome (*Figure 1B*; *Viets et al., 2019*; *Rowley et al., 2019*). However, the molecular nature of these hypothesized buttons is as yet unclear, nor is it clear whether this proposed model could lead to de novo pairing in the absence of some unknown active process that identifies and pairs homologous loci.

Here we turned the 'button' mechanism for somatic homolog pairing into a precise biophysical model that defines parameters for the activities of pairing buttons, informed by observations of pairing dynamics in living cells. Our simulations showed that chromosome-wide pairing can be established through random encounters between specifically interacting buttons that are dispersed across homologous chromosomes at various possible densities using a range of binding energies that are reasonable for protein–protein interactions. Importantly, we found that active processes are not necessary to explain pairing via our model, as all of the interactions necessary for stable pairing are initiated by reversible random encounters that are propagated chromosome-wide. We tested our model and constrained its free parameters by assessing its ability to predict pairing dynamics measured via live imaging. Our model successfully predicted that, once paired, homologous loci remain together in a highly stable state. Furthermore, the model also accurately predicted the dynamics of pairing through the early development of the embryo, as measured by the percentage of nuclei that become paired as development proceeds and by the dynamic interaction of individual loci as they transition from unpaired to paired states. In sum, through an interplay between theory and experiment aimed at probing molecular mechanisms, our analysis provides quantitative data that strongly support a button model as the underlying mechanism of somatic homolog pairing and establishes

the conceptual infrastructure to uncover the molecular identity, functional underpinnings, and regulation of these buttons.

## Results

### Formalizing a button-based polymer model of homologous pairing

Prior studies have suggested that somatic homolog pairing in *Drosophila* may operate via a button mechanism between homologous loci (*AlHaj Abed et al., 2019*; *Erceg et al., 2019*; *Fung et al., 1998*; *Gemkow et al., 1998*; *Rowley et al., 2019*; *Viets et al., 2019*). In this model, discrete regions capable of pairing specifically with their corresponding homologous segments are interspersed throughout the chromosome. To quantitatively assess the feasibility of a button mechanism, we implemented a biophysical model of homologous pairing (*Figure 2*). Briefly, we modeled homologous chromosome arms as polymers whose dynamics are driven by short-range, attractive, specific interactions between homologous loci (buttons) to account for pairing (Materials and methods). These buttons are present at a density ρ along the chromosome and bind specifically to each other with an energy $E_p$. We included short-range, non-specific interactions among (peri)centromeric regions to account for the large-scale HP1-mediated clustering of centromeres (Materials and methods), which may also impact global genome organization inside nuclei (*Rosin et al., 2018*; *Strom et al., 2017*) and thus may affect pairing. As initial conditions for our simulations, we generated chromosome configurations with all centromeres at one pole of the nucleus (a 'Rabl' configuration; *Figure 2—video 1* and *2*), typical of early embryonic fly nuclei (*Dernburg et al., 1996*). To account for the potential steric hindrance of non-homologous chromosomes that could impede pairing, we simulated two pairs of homologous polymers. Note that other polymer models have previously been developed to study homologous pairing but mainly in a meiotic context: *Nicodemi et al., 2008a*; *Nicodemi et al., 2008b* proposed a generic model where homologous chromosomes are constrained to remain parallel and elongated and can interact via non-specific interactions, *Penfold et al., 2012* investigated the role of centromeres and telomeres tethering in yeast on the inter-homolog distances but without accounting for any explicit pairing mechanisms, and *Marshall and Fung, 2016* developed a glue-like model where homologous loci remain attached together when they first meet.

When we monitored the distances between homologous loci in our simulations as a function of time (*Figure 2B*, *Figure 2—figure supplement 1*), qualitatively we observed that this thermodynamic model can lead to the time-progressive pairing of homologous chromosomes (*Figure 2B*) and the gradual intermingling of the two homologous chromosome territories (*Figure 2C*). Pairing in our simulations operates via a stochastic zippering process: once random fluctuations lead to the pairing of one pair of homologous loci, the pairing of nearest-neighbor buttons is facilitated along the lengths of the homologous chromosomes in a zipper-like manner (*Figure 2B*). Full chromosome-wide pairing results from the progression of many zippers that 'fire' at random positions and times along the chromosome, as also previously predicted for meiotic homologous pairing (*Marshall and Fung, 2016*).

We systematically investigated the roles of button density along the genome ρ, of the strength of the pairing interaction $E_p$, and of the initial distance between homologous chromosomes $d_i$ in dictating pairing dynamics (*Figure 2D*). For a given density, there is a critical value of $E_p$ below which no large-scale pairing event occurs independently of the initial conditions (*Figure 2—figure supplement 2A*) since pairing imposes a huge entropic cost for the polymers and thus requires a sufficient amount of energy to be stabilized. Beyond this critical point, higher strengths of interactions and higher button densities lead to faster and stronger pairing (*Figure 2D*, left, center). We also find that the non-specific interactions among (peri)centromeric regions included in our model facilitate pairing, but that such interactions are not strictly necessary (*Figure 2—figure supplement 2D*).

The initial spatial organization of chromosomes also strongly impacts pairing efficiency. When homologous chromosomes are initially far apart, pairing is dramatically slowed and impaired (*Figure 2D*, right) due to the presence of the other simulated chromosomes between them (*Figure 2—figure supplement 2B*). We also observed that our initial chromosome configurations corresponding to a Rabl-like organization (with all centromeres at one pole of the nucleus) promotes pairing by allowing homologous buttons to start roughly aligned (*Marshall and Fung, 2016*;

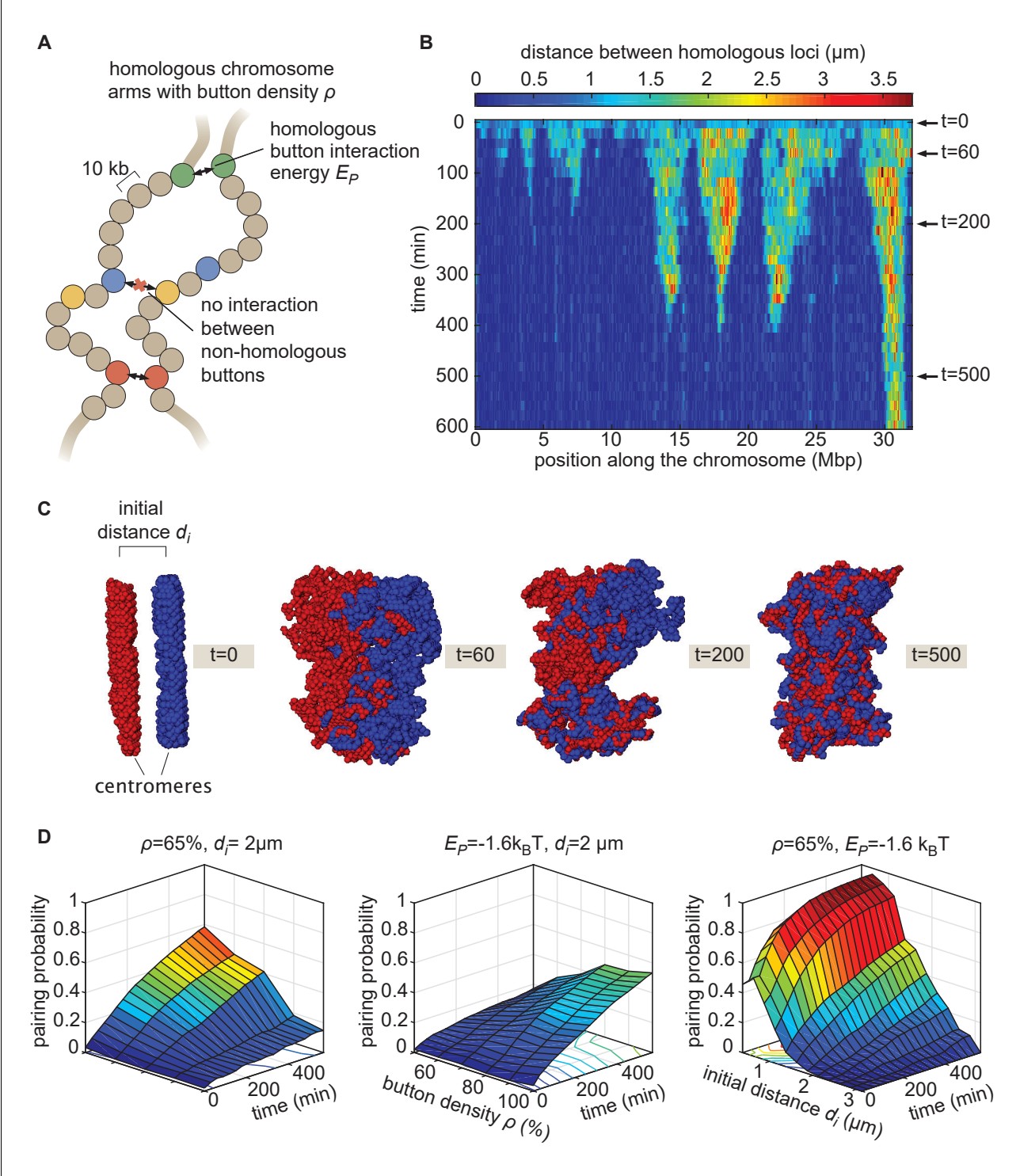

**Figure 2.** The homologous button model. (A) Pairing between homologous chromosomes is assumed to be driven by specific, short-range attractive interactions of strength $E_p$ between certain homologous regions, named buttons. Each 10-kb monomer in the simulation corresponds to one locus. (B) Kymograph of the time evolution of the distances between homologous regions predicted by the model in one representative simulated stochastic trajectory for a button density of $\rho = 65\%$, an interaction strength of $E_p = -1.6k_BT$, and an initial distance $d_i = 1\mu m$. See *Figure 2—figure supplement 1* for other examples for various $d_i$ values. (C) Snapshots of the pair of homologous chromosomes at various time points along the simulation in (B) (see also *Figure 2—video 1* and *2*). (D) Predicted average pairing probability between euchromatic homologous loci (considered as paired if their

*Figure 2 continued on next page*

*Figure 2 continued*

relative distance $\leq 1\mu m$) as a function of time and of the strength of interaction $E_p$ (left), the button density $\rho$ (center), and the initial distance $d_i$ between homologous chromosomes (right).

The online version of this article includes the following video and figure supplement(s) for figure 2:

**Figure supplement 1.** Simulated time evolution of distance between homologous loci.
**Figure supplement 2.** Properties of the button model.
**Figure supplement 3.** Large-scale correlations in pairing probabilities.
**Figure supplement 4.** The combinatorial and large button models.
**Figure 2—video 1.** Polymer simulations of homologous pairing.
https://elifesciences.org/articles/64412#fig2video1
**Figure 2—video 2.** Polymer simulations of homologous pairing.
https://elifesciences.org/articles/64412#fig2video2

*Penfold et al., 2012*; *Nicodemi et al., 2008b*; *Figure 2—figure supplement 2F*). Taken together, these systematic analyses of model parameters support the view that the homologous button model is compatible with pairing.

As an alternative model, we asked whether buttons that interact non-specifically could also explain somatic pairing. We simulated the dynamics of polymers having such non-specific buttons and never observed significant chromosome-wide pairing (*Figure 2—figure supplement 2C,E*). These results are complementary to previous works in which we showed that the weak, non-specific interactions between epigenomic domains that drive TAD and compartment formation in *Drosophila* (*Ghosh and Jost, 2018*; *Jost et al., 2014*) cannot establish and maintain stable pairing by themselves (*Pal et al., 2019*). Thus, in addition to button density, interaction strength, and initial organization of chromosomes, a key mechanism for pairing is the specificity of preferential interactions between homologous regions.

## Live imaging reveals homologous pairing dynamics

The button model in *Figure 2* makes precise predictions about pairing dynamics at single loci along the chromosome. To inform the parameters of the model and to test its predictions, it is necessary to measure pairing dynamics in real time at individual loci of a living embryo. To do so, we employed the MS2/MCP (*Bertrand et al., 1998*) and PP7/PCP (*Chao et al., 2008*) systems for labeling nascent transcripts. Here, each locus contains MS2 or PP7 loops that can be visualized with distinct colors in living embryos (*Fukaya et al., 2016*; *Lim et al., 2018*; *Chen et al., 2018*; *Garcia et al., 2013*). Specifically, we designed transgenes encoding MS2 or PP7 loops under the control of UAS (*Brand and Perrimon, 1993*) and integrated them at equivalent positions on homologous chromosomes (*Figure 3A*). Activation of transcription with GAL4 creates nascent transcripts encoding the MS2 or PP7 stem loops, each of which can be directly visualized by maternally providing fluorescently labeled MCP (MCP-mCherry) or PCP (PCP-GFP) in the embryo. The accumulation of fluorescent molecules on nascent transcripts was detected via laser-scanning confocal microscopy, providing relative three-dimensional positions of actively transcribing chromosomal loci in living *Drosophila* embryos (*Lim et al., 2018*; *Chen et al., 2018*).

We focused on embryos that had completed the maternal-to-zygotic transition and began to undergo gastrulation at approximately 2.5–5 hr after embryo fertilization, when pairing begins to increase substantially (*Fung et al., 1998*). We integrated transgenes into two genomic locations on chromosome two at polytene positions 38F and 53F, and analyzed embryos with MS2 and PP7 loops at the same positions on homologous chromosomes in order to monitor pairing (*Figure 3B*, top; *Figure 3—video 1* and *2*; Materials and methods). As a negative control, we imaged embryos in which loops were integrated at two different positions on homologous chromosomes (MS2 at position 38F and PP7 at position 53F), where we expect no pairing between transgenes (*Figure 3B*, middle; *Figure 3—video 3*). Finally, as a positive control for the spatial colocalization of MS2 and PP7 loops, we analyzed embryos where MS2 and PP7 loops were interlaced in a single transgene (*Wu et al., 2014*; *Chen et al., 2018*) on one chromosome at polytene position 38F (*Figure 3B*, bottom; *Figure 3—video 4*). For each case, we imaged multiple embryos for 30–60 min, and used custom MATLAB scripts to determine the relative 3D distances between chromosomal loci over time (*Figure 3—figure supplement 1*; Materials and Methods).

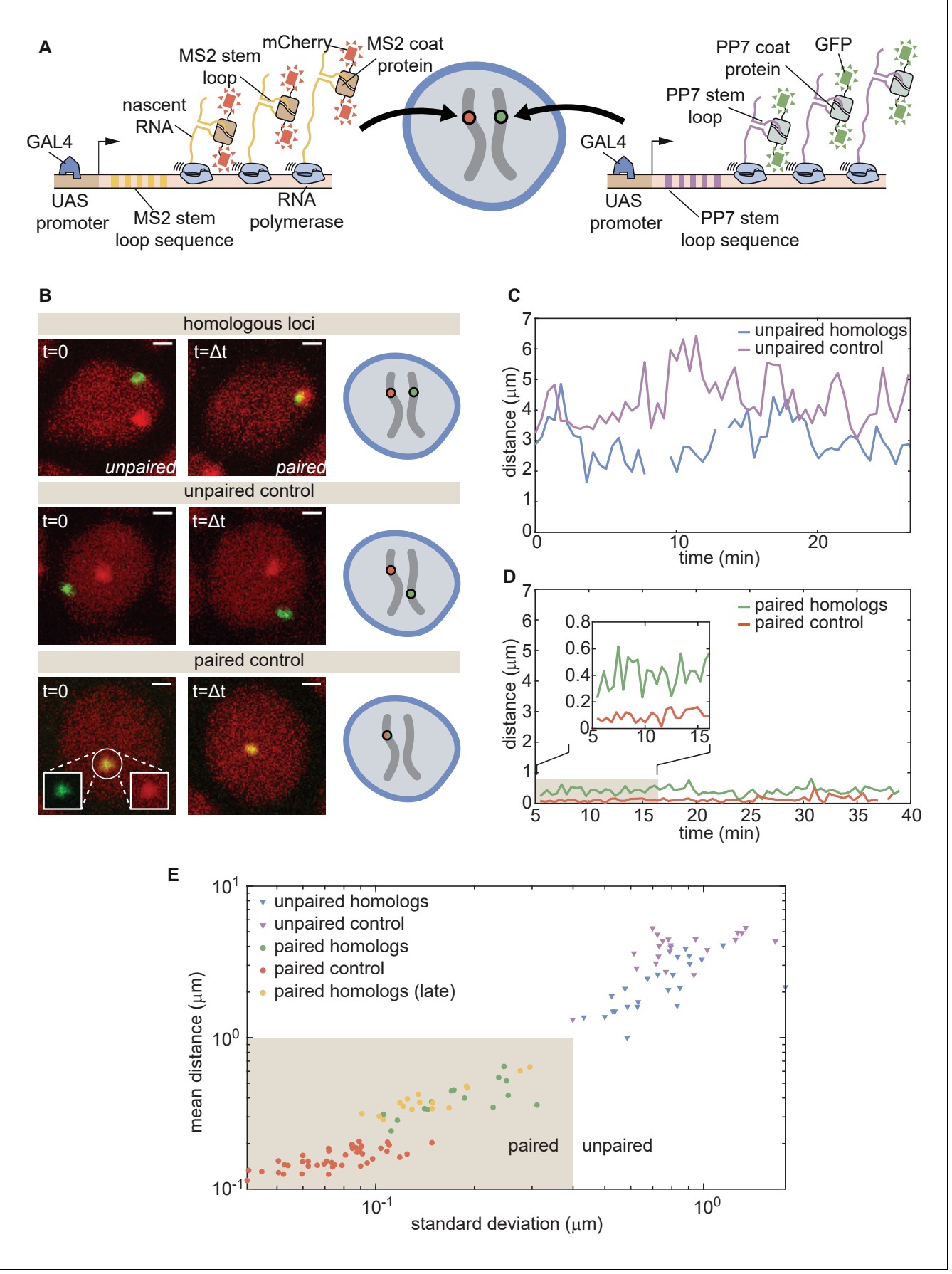

**Figure 3.** Live imaging of chromosomal loci provides dynamic single-locus spatiotemporal information about somatic homolog pairing. (**A**) Schematic of the MS2 and PP7 nascent mRNA labeling scheme for live imaging of homologous loci. Expression of the stem loops is driven by UAS under the control of a GAL4 driver. (**B**) Snapshots at two time points from homologous chromosomal loci with one allele tagged with MS2 and one allele tagged with PP7 (top), negative controls consisting of non-homologous loci labeled with MS2 and PP7 (middle), and positive controls corresponding to a single reporter containing interlaced MS2 and PP7 stem loops on the same chromosome (bottom). Scale bars represent 1 μm. See also *Figure 3—videos 1*, *2*, *3*, *4*. (**C**) Representative traces of the dynamics of the distance between imaged loci for unpaired homologous loci and the negative control showing how both loci pairs have comparable distance dynamics. (**D**) Representative traces of the dynamics of the distance between imaged loci for paired homologous loci and the positive control demonstrating how the distance between paired loci is systematically higher than the control. (**E**) Mean and standard deviation (SD) of the distance between reporter transgenes, where each data point represents a measurement over the length of time that the loci were imaged (ranging from approximately 10–50 min, depending on the duration of the movie and the length of time that a nucleus remained in the field of view, see *Figure 3—figure supplement 1*). The shaded region indicates the criterion used to define whether homologs are paired (mean distance < 1.0 μm, SD < 0.4 μm) based on the distribution of points where homologs were qualitatively assessed as paired (yellow and green points). The online version of this article includes the following video and figure supplement(s) for figure 3:

**Figure supplement 1.** Experimental dynamics of inter-homolog distances used in *Figure 3*.

**Figure supplement 2.** Homologous chromosome reporters inserted at the 53F genomic location.

**Figure 3—video 1.** Representative confocal movie of a live *Drosophila* embryo (cell cycle 14 to gastrulation) in which MS2 and PP7 loops are integrated at equivalent positions on homologous chromosomes.

https://elifesciences.org/articles/64412#fig3video1

**Figure 3—video 2.** Representative confocal movie of a live *Drosophila* embryo (roughly 4.5 hr old) in which MS2 and PP7 loops are integrated at equivalent positions on homologous chromosomes.

https://elifesciences.org/articles/64412#fig3video2

**Figure 3—video 3.** Representative confocal movie of a live *Drosophila* embryo (roughly 5.5 hr old) in which MS2 and PP7 loops are integrated at various positions on homologous chromosomes (MS2 at position 38F and PP7 at position 53F) where we expect no pairing between transgenes.

https://elifesciences.org/articles/64412#fig3video3

**Figure 3—video 4.** Representative confocal movie of a live *Drosophila* embryo (cell cycle 14) in which MS2 and PP7 loops were interlaced in a single transgene on one chromosome at polytene position 38F to act as a positive control for pairing.

https://elifesciences.org/articles/64412#fig3video4

In embryos with both PP7 and MS2 transgenes integrated at polytene position 38F (*Figure 3B*, top), the majority of nuclei could be qualitatively classified into one of two categories. In 'unpaired' nuclei, homologous loci were typically separated by > 1 μm with large and rapid changes in inter-homolog distances (e.g. *Figure 3C*, blue), with a mean distance of 2.2 μm and standard deviation (SD) of 1.2 μm averaged over 30 nuclei. The measured mean distance between homologous loci was comparable within error, though systematically smaller, than the mean distance between loci in the negative control, where transgenes were integrated at non-homologous positions (*Figure 3C*, red, mean distance = 4.0 μm, SD = 1.3 μm, n = 21 nuclei). In contrast, in 'paired' nuclei, homologous loci remained consistently close to one another over time, with smaller dynamic changes in inter-homolog distance (*Figure 3D*, blue, mean distance = 0.4 μm, SD = 0.3 μm, n = 25 nuclei). Interestingly, while the diffraction-limited signals produced from homologous loci occasionally overlapped in paired nuclei, their average separation was systematically larger than that of the positive-control embryos carrying interlaced MS2 and PP7 loops (*Figure 3D*, red, mean distance = 0.2 μm, SD = 0.1 μm, n = 44 nuclei). This control measurement also constitutes a baseline for the experimental error of our quantification of inter-homolog distances (*Chen et al., 2018*). Our measurements thus confirmed previous observations of transgene pairing in the early embryo in which signals from paired loci maintained close association but did not completely coincide over time (*Lim et al., 2018*). Notably, of 38 nuclei qualitatively scored as having paired homologs, we never observed a transition back to the unpaired state over a combined imaging time of more than 8 hr. Embryos with PP7 and MS2 transgenes integrated in homologous chromosomes at polytene position 53F showed comparable dynamics of inter-homolog distances for nuclei in unpaired and paired states (*Figure 3—figure supplement 2A,B*). Thus, somatic homolog pairing is a highly stable state characterized by small dynamic changes in the distance between homologous loci.

Our assessment thus far has been based on a qualitative definition of pairing. In order to devise a more stringent quantitative definition of homologous pairing, we measured inter-transgene distances for homologous loci as well as for the unpaired and paired controls throughout gastrulation. We also included measurements from older embryos (~11–12 hr after fertilization) using the driver *R38A04-GAL4* (*Jenett et al., 2012*) to express the transgenes in epidermal cells, where pairing is

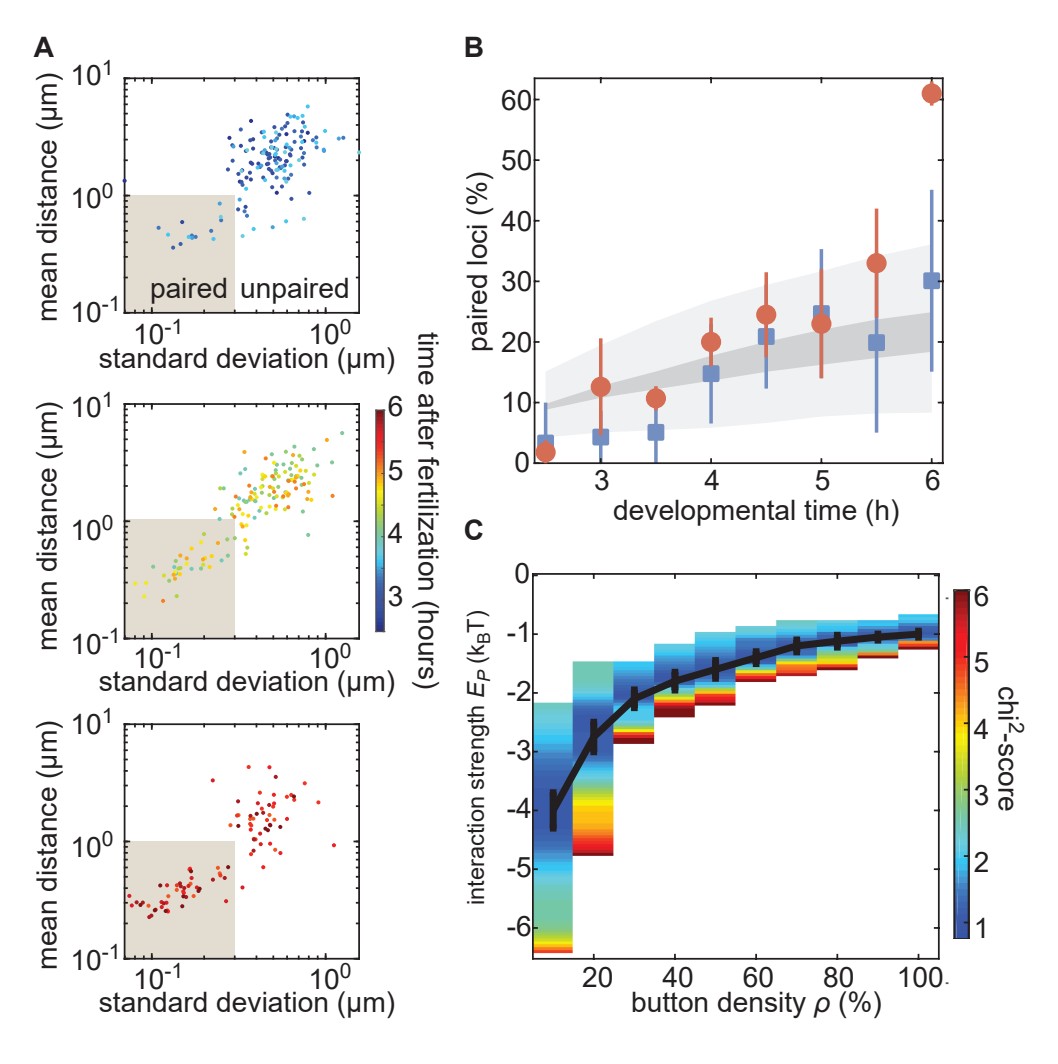

**Figure 4.** The homologous button model recapitulates the observed developmental dynamics of pairing. (A) Mean and SD of the separation of each pair of transgenes integrated at position 38F imaged in a single embryo over 6 h of development. Each data point represents a single nucleus over a 10-min time window, revealing the increase in the fraction of paired loci as development progresses. Data are separated into three plots for ease of visualization. (B) Nuclei from each time point were scored as "paired" if they fell within the shaded box in (A). Data were taken from three embryos each for transgenes at 38F (red) and 53F (blue) with error bars representing the standard error of the mean. For each button density ρ, we fitted the experimental pairing dynamics (*Figure 4— figure supplement 2A*). Gray shading provides the envelope of the best predictions obtained for each ρ (dark gray) and its SD (light gray). (C) Phase diagram representing, as a function of ρ, the value of $E_p$ (black line) that leads to the best fit between predicted and experimental developmental pairing dynamics. The predicted pairing strength is weaker than observed in the parameter space above the line, and stronger than observed below the line. Error bars represent the uncertainties on the value of $E_p$ that minimizes the chi$^2$-score at a given ρ value. The online version of this article includes the following figure supplement(s) for figure 4:

**Figure supplement 1.** Comparison of our pairing data to previous results.
**Figure supplement 2.** Establishing values for initial distance $d_i$ between homologous chromosomes via chromosome painting.
**Figure supplement 3.** Inference of developmental pairing dynamics.

expected to be widespread (*Fung et al., 1998*; *Gemkow et al., 1998*). We measured the mean and SD of the inter-transgene distance for each nucleus over ~10–50 min. From these data, we established a quantitative and dynamic definition of somatic homolog pairing based on a mean

distance < 1.0 µm and a corresponding standard deviation < 0.4 µm (*Figure 3E*, shaded region). By this definition, we considered paired 100% of nuclei that we had qualitatively scored as such, but excluded all nuclei scored as unpaired. As expected, this definition also scored 100% (15/15) of the tracked nuclei from older embryos as paired. Data for paired nuclei from early versus late embryos were in close agreement (*Figure 3E*, yellow), suggesting that pairing observed in early embryos is representative of pairing during later stages of development.

We next analyzed the progression of pairing through the first 6 hr of development in single embryos carrying MS2 and PP7 transgenes in homologous chromosomes at positions 38F and 53F. To accomplish this goal, we collected data for short (~10 min) intervals every 30 min from 2.5 hr to 6 hr of development, and analyzed inter-homolog distances as outlined above. We then plotted the mean of this distance as a function of its SD for each nucleus analyzed at each time point to create a dynamic assessment of somatic homolog pairing over developmental time. As expected, we detected an overall decrease in mean inter-homolog distance and its SD as development progressed (*Figure 4A*, *Figure 3—figure supplement 2C*). To directly compare our analysis to prior studies, we binned nuclei into paired and unpaired states based on their mean and SD as defined in *Figure 3E* and plotted the percentage of paired nuclei at each developmental time point (*Figure 4B*). Consistent with previous literature (*Fung et al., 1998*; *Gemkow et al., 1998*), we observed a steady increase in the proportion of paired nuclei (*Figure 4B*); however, by our dynamical definition of pairing, the percentage of nuclei that are paired is systematically lower at most time points than results using DNA-FISH (*Figure 4—figure supplement 1*). This disagreement likely reflects differences between the classic, static definition of pairing based on overlapping DNA-FISH signals in the one snapshot accessible by fixed-tissue measurements as opposed to our dynamics-based definition, which demands that loci be paired over several consecutive frames. In sum, we have demonstrated that our system captures the progression of somatic homolog pairing over developmental time, making it possible to contrast theoretical predictions and experimental measurements.

## Constraining the button model using dynamical measurements of pairing probability

Our button model predicts that the fraction of paired loci as a function of time depends on three parameters: the initial separation between homologous chromosomes $d_i$, the density of buttons along the chromosome ρ, and the button–button interaction energy $E_p$ (*Figure 2D*). As an initial test of our model, and to constrain the values of its parameters, we sought to compare model predictions to experimental measurements of the fraction of paired loci over developmental time.

Due to the still unknown molecular identity of the buttons, it was impossible to directly measure the button density and the button–button interaction energy. However, the initial separation between chromosomes $d_i$ can be directly estimated using chromosome painting (*Ried et al., 1998*; *Beliveau et al., 2012*). To make this possible, we used Oligopaint probes (*Beliveau et al., 2012*) targeting chromosome arms 2L and 2R to perform chromosome painting on embryos ~ 130 min after fertilization, corresponding to the beginning of cell cycle 14 (*Foe, 1989*; *Figure 4—figure supplement 2A*). The resulting distribution of distances between homologous chromosome territories was well described by a simple Gaussian distribution for distances greater than 1 µm, roughly corresponding to the distance required to resolve two separate chromosome territories (*Figure 4—figure supplement 2B*, red line.)

We next investigated whether the button model quantitatively reproduced the pairing dynamics observed during development for reasonable values of the button density and the button–button interaction energy. We ran a series of simulations for various values of button density ρ (from 10% to 100%) and strength of interaction $E_p$ (from $-0.5k_BT$ to $-5k_BT$) starting from values for the initial distance between homologous chromosomes $d_i$ drawn from the inferred Gaussian distribution from our Oligopaint measurements (*Figure 4—figure supplement 2B*, black line). For each parameter set, we monitored pairing dynamics as a function of developmental time and computed the average probability for a locus to be paired (*Figure 4—figure supplement 3A*, black points) using the same criterion as in *Figure 3E*. By minimizing a chi²-score (*Figure 4—figure supplement 3B*) between the predictions and the experimental pairing probability (Materials and methods), we inferred, for each button density, the strength of interaction that best fits the data (*Figure 4C*). Interestingly, the goodness of fit was mainly independent of button density (*Figure 4—figure supplement 3C*):

denser buttons require less strength of interaction to reach the same best fit (black line in *Figure 4C*).

The inferred developmental dynamics quantitatively recapitulated the experimental observations for both investigated loci at the majority of time points analyzed (*Figure 4B*) for any choice of parameters given by the curve in *Figure 4C*. At our initial time point of 2.5 hr, we predict pairing to be slightly higher than observed for position 38F. This disagreement could reflect an underestimate of the initial distance $d_i$ in our simulations at distances less than 1 µm, which corresponds to the resolution limit of our Oligopaint-based measurements (*Figure 4—figure supplement 2B*), or that homolog pairing is not yet stable early in *Drosophila* development (*Lim et al., 2018*). We also find that our simulations did not predict the large increase in pairing observed for 38F between 5.5 hr and 6 hr (*Figure 4B*), which may be a consequence of the proximity of 38F to the highly paired histone locus body (*Hiraoka et al., 1993*; *Fung et al., 1998*). In sum, the button model recapitulates the observed average pairing dynamics for a wide range of possible button densities coupled with interaction energies that are consistent with protein-DNA interactions.

## Parameter-free prediction of individual pairing dynamics

The fit of our button model to the fraction of paired loci during development in living embryos (*Figure 4B*) revealed a dependency between the interaction strength $E_p$ and the buttondensity ρ (*Figure 4C*). As a critical test of the model's predictive power, we sought to go beyond averaged pairing dynamics and used the model to compute the pairing dynamics of individual loci. As can be seen qualitatively in the kymographs predicted by the model (*Figure 2B*, *Figure 2—figure supplement 1*), pairing spreads rapidly (within tens of minutes) from the buttons that constitute the initial points of contact along the chromosome. As a result, the button model predicts that homologous loci undergo a rapid transition to the paired state as the zippering mechanism of pairing progression moves across the chromosome.

To quantify the predicted pairing dynamics of homologous loci, we collected single-locus traces containing individual pairing events from our simulations (*Figure 5A*, top), which we defined as traces in which the inter-homolog distance drops below 0.65 µm for at least 4 min (Materials and methods). For traces corresponding to each set of simulations with various values of ρ and $E_p$ (*Figure 5B*), we calculated the median dynamics of inter-homolog distances around the pairing event. Across many values of ρ and $E_p$, the medians of the predicted trajectories leading up to the pairing event were very similar, with inter-homolog distances decreasing rapidly from 1 to 2 µm to below 0.65 µm at an accelerating rate over the course of 10–20 min (*Figure 5C–F*). However, we do observe subtle differences in this pre-pairing stage: for a given button density, a stronger interaction energy $E_p$ leads to a faster approach of the homologs (*Figure 5—figure supplement 2A*). The nearly independence of this first period of the pairing dynamics with respect to ρ and $E_p$ suggests that the initial approach of homologous loci is mainly diffusion limited, while there is a slight acceleration of pairing for stronger interaction energies due to an enhanced zippering effect (*Figure 5—figure supplement 2B*).

In contrast to the initial pairing dynamics, varying model parameter values had a clear effect on the distance dynamics that followed the pairing event (*Figure 5—figure supplement 2A*). Specifically, simulations with a weak $E_p$ led to a slow increase in inter-homolog distances as time progressed (*Figure 5C*, red), consistent with unstable pairing events. Conversely, simulations with a strong $E_p$ were associated with tight pairing of homologous loci following the pairing event, with inter-homolog distances stably maintained around 130 nm, close to the spatial resolution of the model (*Figure 5D*, green). Notably, the values of ρ and $E_p$ that best fit the averaged temporal evolution of the fraction of paired loci over development (*Figures 4* and *5B*) all led to similar predictions for the median inter-homolog distance dynamics associated with pairing events. These traces converged to a stable long-term median inter-homolog distance of ~0.5 µm (*Figure 5E,F*), which is nearly identical to the experimentally determined distance of ~0.44 µm between homologous loci in stably paired nuclei (compare the colored and black lines in *Figure 5E,F*). Our results thus suggest that the slow dynamics of the pairing probability observed during development (*Figure 4*) and the dynamics of inter-homolog distance after a pairing event (*Figure 5*) are strongly correlated.

We then compared our simulated traces to experimental observations of pairing events in nuclei of live embryos. Among the movies that we monitored, we captured 14 pairing events matching the

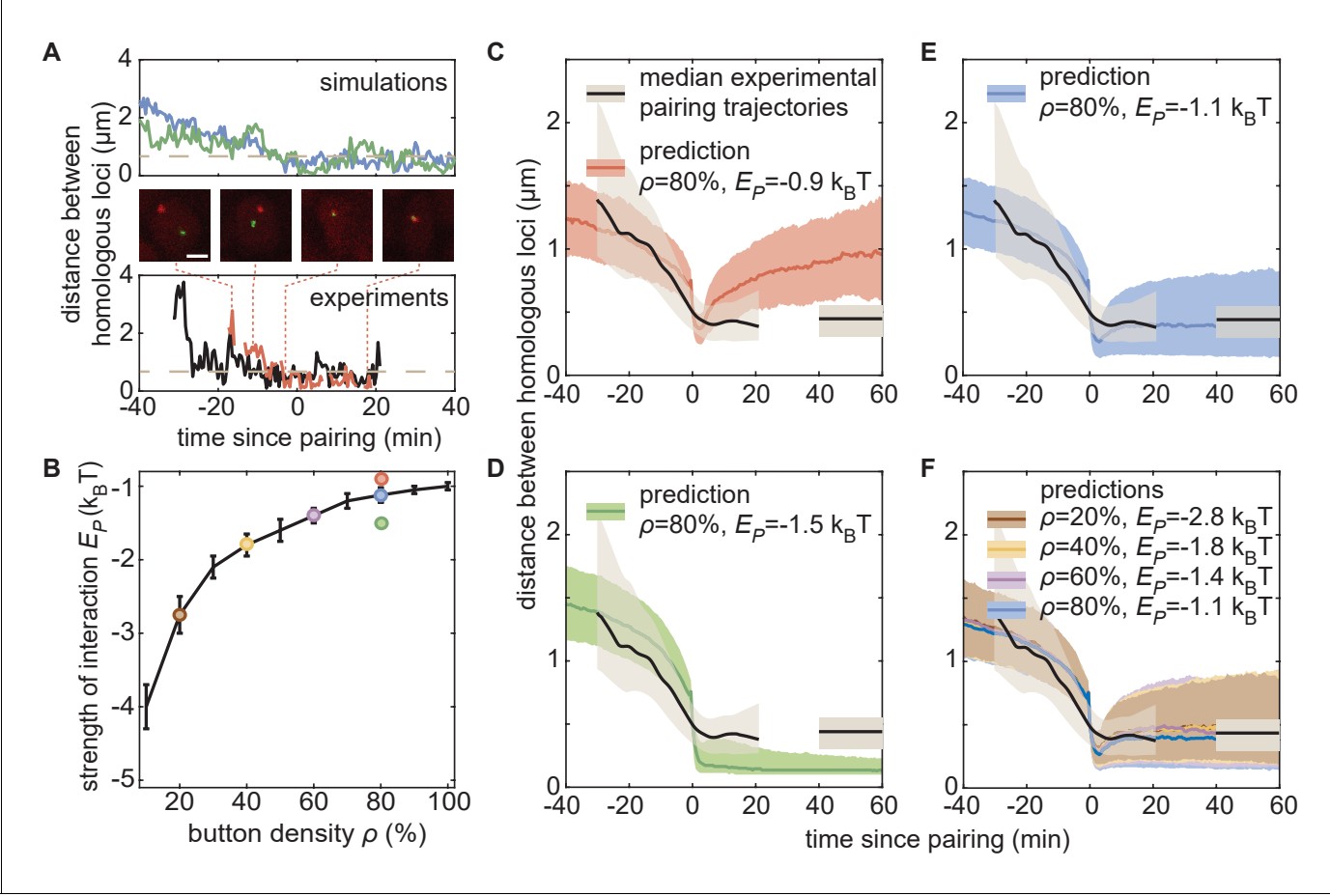

**Figure 5.** The homologous button model predicts individual pairing dynamics. (**A**) Examples of simulated (top) and experimental (bottom) pairing trajectories showing rapid transitions from the unpaired to the paired state. Simulations were carried out using ρ = 50% and $E_p = -1.75 k_B T$. See also *Figure 5—figure supplement 1*. Scale bar is 2 μm. Dotted line in each graph represents our distance threshold for aligning pairing traces defined as the time point where the inter-homolog distance decreases below 0.65 μm for at least 4 min. (**B**) Parameter range inferred from pairing probability dynamics in *Figure 4* (black line), and parameters used for the simulations in (**C–F**) (color points). (**C–F**) Median pairing dynamics obtained from individual pairing trajectories detected during our experiments (black lines) and simulations (colored lines). Traces are centered at the time of pairing (time = 0) as in (**A**). The long-term, experimentally measured inter-homolog distance is plotted as a straight black line at the right of each panel. The interquartile range of the distributions of distances between homologous loci are indicated by the shaded regions (n = 14 nuclei for experiments; n > 10,000 traces for simulations). Note that the experimental data were processed (*Figure 5—figure supplement 1C*) to smooth out the effect of small statistics (*Figure 5—figure supplement 2C*).

The online version of this article includes the following video and figure supplement(s) for figure 5:

**Figure supplement 1.** Experimental individual pairing dynamics.

**Figure supplement 2.** Impact of $E_p$ and small statistics on individual pairing dynamics.

**Figure 5—video 1.** Representative distance trajectory of two loci denoted as 'pairing' (plotted in the black experimental trace in *Figure 5A*) showing a rapid transition from large distances at earlier time points to smaller distances at later time points.

https://elifesciences.org/articles/64412#fig5video1

criteria of initial large inter-homolog distances that drop below 0.65 μm for at least 4 min (*Figure 5A*; *Figure 5—figure supplement 1*; *Figure 5—video 1*). We aligned each of these pairing events using the same approach as with the simulated data described above and calculated the smoothed median dynamics of inter-homolog distances around the pairing event (*Figure 5C–F*, black lines, see also *Figure 5—figure supplement 1C*). The pre-pairing dynamics were fully compatible with model predictions, with a rapid decrease in inter-homolog distances over 10–20 min (*Figure 5C–F*, see also *Figure 5—figure supplement 2C*). Furthermore, the experimental post-pairing dynamics in inter-homolog distance were closely recapitulated (*Figure 5F*) by the predictions

made using parameters that best fit the pairing probability over developmental time (*Figure 5B*). In sum, simulations of chromosomal behavior based on a button model with a defined set of parameters quantitatively recapitulate experimental observations of pairing events at individual loci, of stably paired homologs following a pairing event, and of the global progression of pairing dynamics over developmental time.

## Discussion

Since its discovery by Nettie Stevens over 100 years ago (*Stevens, 1908*), somatic homolog pairing has represented a fascinating puzzle for geneticists and cell biologists alike. The dissection of the molecular origins of somatic pairing presents a tractable case study to further our understanding of the 3D organization of chromosomes and the functional consequences of interactions among otherwise distant DNA loci. However, despite decades of research, the molecular mechanisms underlying somatic homolog pairing have remained elusive (*Joyce et al., 2016*). In this paper, we augmented the emerging button-based cartoon model of somatic homolog pairing by turning it into a precise theoretical model that makes quantitative and testable predictions of pairing dynamics as a function of the density of buttons throughout the chromosome and the specific interaction energy between buttons.

To assess the feasibility of this button model, we used it to predict chromosomal dynamics and then tested those predictions experimentally by tracking pairing dynamics at individual chromosomal loci in living embryos. Simulations predicted rapid transitions from unpaired to paired states resulting from a 'zippering' effect across the chromosomes where buttons that become paired via random encounters promote and stabilize pairing of adjacent buttons (*Figure 2B*). The model predicts that the spread of pairing from button to button along the length of the chromosome ultimately leads to the formation of paired homologous chromosomes that remained associated throughout the remainder of the simulation. This process gives rise to significant large-scale correlations between the pairing probabilities of distant loci, spreading over large genomic distances (~Mbp) as global pairing progresses during development (*Figure 2—figure supplement 3*).

The notion of zippering was previously proposed in a classical model of somatic pairing by Ed Lewis (*Duncan, 2002*) although, in his model, pairing initiates exclusively from the centromeres and propagates out toward the telomeres. In contrast, our data shows that pairing initiates randomly at multiple chromosomal positions. In this way, our model supports a prior study that used DNA-FISH on fixed embryos to demonstrate that pairing initiates at independent loci along the chromosome (*Fung et al., 1998*), and is consistent with polymer modeling that also suggests zippering as a possible mechanism for meiotic pairing (*Marshall and Fung, 2016*). While our experimental validation of the button model is currently limited to the tracking of a single pair of homologous loci at a time, the simultaneous live imaging of several loci would enable a more complete test of the collective, large-scale dynamics emerging from the predicted zippering process. Recent progress in the labeling of multiple loci may make this challenge possible in the coming years (*Chen et al., 2018*).

In tracking pairing dynamics through early development in living embryos, we found quantitative agreement with the button model predictions: the transition from an unpaired to a paired state is a rapid event that occurs in just a few minutes (*Figure 5A*), and paired chromosomal loci remain stably paired over the observation time of our experiments, up to 45 min (data not shown). Overall, the close quantitative agreement between observation and theory validates the button model as a mechanism that supports the initiation and maintenance of somatic homolog pairing. Furthermore, our measurements constrain the range of possible values of the button density and interaction energy (*Figure 4C*).

Two caveats may be considered in interpreting our analysis. First, our method of tracking homologous loci in living embryos relies on visualizing nascent RNAs generated from transgenes (Materials and methods) rather than direct observations of DNA or DNA-binding proteins. While nascent RNAs provide a robust and convenient signal for the position of the underlying DNA (*Lim et al., 2018*; *Chen et al., 2018*), the method limits us to examining the behavior of transcriptionally active loci, which could behave differently from silent chromatin. In addition, our analysis could overestimate inter-homolog distances in paired nuclei if, for example, nascent RNA molecules from separate chromosomes are prevented from intermixing (*Fay and Anderson, 2018*). Second, our simulations do not account for complex behaviors of the genome that take place during development and that may

also influence pairing dynamics and stability, including cell-cycle progression and mitosis (*Foe, 1989*), establishment of chromatin states and associated nuclear compartments (*Sexton et al., 2012*; *Yuan and O'Farrell, 2016*; *Hug et al., 2017*; *Ogiyama et al., 2018*), and additional nuclear organelles such as the histone locus body (*White et al., 2011*; *Liu et al., 2006*). Further testing and refinement of our theoretical and molecular understanding of somatic homolog pairing will require new approaches to incorporate the potential influences of these genomic behaviors in a developmental context.

A previous analysis of pairing and transvection in living embryos focused on the blastoderm phase, coinciding with the earliest developmental time points in our analysis, and found that inter-homolog interactions were generally unstable at that time (*Lim et al., 2018*). Thus, the embryo appears to transition from an early state where pairing is not stable prior to cellularization to one that supports stable pairing at later time points of development. Prior studies have postulated changes in cell-cycle dynamics (*Fung et al., 1998*; *Gemkow et al., 1998*), chromatin states (*Bateman and Wu, 2008*), or proteins that promote or antagonize pairing (*Joyce et al., 2012*; *Bateman et al., 2012a*; *Hartl et al., 2008*; *Rowley et al., 2019*) as potentially mediating a shift to stable pairing during the maternal-to-zygotic transition that occurs during blastoderm cellularization. Our data suggest that these changes mediate their effect on pairing by directly or effectively modulating button activity.

What is the molecular nature of the buttons? Prior studies based on Hi-C methods reported that the *Drosophila* genome contains small (a few kbps) distinct regions or peaks of tight pairing between homologs distributed with a typical density of 60–70% throughout the chromosome, which could represent pairing buttons (*AlHaj Abed et al., 2019*; *Erceg et al., 2019*; *Rowley et al., 2019*). Given such a button density and our experimental observations, our model predicts that a specific interaction energy between buttons would be ~1–2 $k_B$T (*Figure 4C*), a value consistent with both typical protein–protein interactions (*Phillips et al., 2012*) and with electrostatic interactions between homologous DNA duplexes (*Kornyshev and Leikin, 2001*; *Gladyshev and Kleckner, 2014*).

Recently, an analysis of ectopically induced pairing in vivo by *Viets et al., 2019* found that relatively large chromosomal segments (~100 kbp) are required to promote pairing, consistent with our model prediction that a region must contain enough 'small' buttons (or tight-pairing regions) at a given interaction strength to become paired (*Figure 4C*). Moreover, two studies independently found enrichment for DNA-binding architectural and insulator proteins in tight-pairing regions (*AlHaj Abed et al., 2019*; *Rowley et al., 2019*), suggesting a potential role for these proteins in button function. In support of this view, *Viets et al., 2019* observed that genomic regions amenable to pairing are enriched in clusters of insulator proteins, and previous works on the incorporation of insulator sequences into transgenes showed that these sequences can stabilize pairing and transvection (*Lim et al., 2018*; *Fujioka et al., 2016*; *Piwko et al., 2019*). Notably, our analysis revealed a requirement for some degree of specificity between homologous buttons (*Figure 2—figure supplement 4A,B*), since simulations of non-specific interactions between buttons did not result in robust pairing (*Figure 2—figure supplement 2E*). Perhaps a 'code' of interactions between unique combinations of insulators and architectural proteins (*AlHaj Abed et al., 2019*; *Rowley et al., 2019*) conveys the necessary specificity between homologous buttons for efficient pairing (*Figure 2—figure supplement 4A,C*). Another complementary possibility is that buttons may form large self-interacting pairing units or specific microcompartments along the genome (*Figure 2—figure supplement 4D,E*), potentially overlapping with the segmentation of the genome into TADs (*Viets et al., 2019*).

While somatic homolog pairing is widespread in *Drosophila* and other Dipterans, it is curious that pairing of homologous sequences is rare in the somatic cells of other diploid species. It is possible that the sequences and proteins that underlie buttons are unique to Dipterans and are not present on the chromosomes of other species, perhaps due to the diversity of architectural proteins carried in the *Drosophila* genome (*Cubeñas-Potts and Corces, 2015*). Alternatively, chromosomes of other species may have the capacity to pair through encoded buttons, but are prevented from doing so by the functions of proteins that antagonize pairing, such as the condensin II complex (*Hartl et al., 2008*; *Joyce et al., 2012*; *Rowley et al., 2019*). However, most other diploid species do show a capacity to pair homologous chromosomes during the early stages of meiosis, and polymer models similar to ours have been proposed as potential mechanisms for meiotic pairing (*Marshall and Fung, 2016*; *Penfold et al., 2012*; *Nicodemi et al., 2008b*). While it is possible that meiotic pairing could be mediated via buttons similar to those postulated here (*Marshall and Fung, 2016*),

important differences appear to exist in the progression of meiotic pairing relative to somatic pairing, such as the highly dynamic and unstable associations between homologous loci (*Ding et al., 2004*) and rapid meiotic prophase chromosome movements (*Lee et al., 2012*) that have been observed in yeast, as well as unique chromosomal regions called pairing centers in *Caenorhabditis elegans* (*MacQueen et al., 2005*; *Phillips et al., 2005*). Therefore, multiple molecular mechanisms may accomplish the goal of aligning homologous chromosomes in different cellular contexts.

Importantly, our biophysical model of the otherwise cartoon-like button model coupled with quantitative live-cell imaging of pairing dynamics establishes a foundational framework for uncovering the parameters of button density and binding energy underlying somatic homolog pairing. In the future, we anticipate that our model will be instrumental in identifying and characterizing candidate button loci and in determining how these parameters are modulated in the mutant backgrounds that affect pairing (*Bateman et al., 2012b*; *Joyce et al., 2012*; *Hartl et al., 2008*; *Gemkow et al., 1998*). For example, titration of candidate pairing factors such as specific insulator proteins may challenge the role of the strength of interactions in maintaining a proper global level of pairing as predicted by the button model (*Figure 2D*, left; *Figure 2—figure supplement 2A*). Deletion of buttons at specific loci may also help dissect the role of button density (*Figure 2D*, center) and the propagation of local perturbations to distal loci (*Figure 2—figure supplement 3B*). Thus, our study significantly advances our understanding of the century-old mystery of somatic homolog pairing and provides a theory-guided path for uncovering its molecular underpinnings.

## Materials and methods

### The homologous button model

We modeled two pairs of homologous chromosome arms as semi-flexible self-avoiding polymers. Each chromosome consists of N = 3200 beads, with each bead containing 10 kbp and being of size *b* nm. The four polymers moved on a face-centered-cubic lattice of size $L_x$ x $L_y$ x $L_z$ under periodic boundary conditions to account for confinement by other chromosomes. Previously, we showed that TAD and compartment formation may be quantitatively explained by epigenetic-driven interactions between loci sharing the same local chromatin state (*Jost et al., 2014*; *Ghosh and Jost, 2018*). However, such weak interactions cannot lead to global homologous pairing (*Pal et al., 2019*). Here, to simplify our model, we neglect these types of interactions (whose effects are mainly at the TAD scale) to focus on the effect of homolog-specific interactions. However, we do consider HP1-mediated interactions between (peri)centromeric regions that are thought to impact the global large-scale organization inside nuclei (*Figure 2—figure supplement 2D*; *Strom et al., 2017*).

Homologous pairing was modeled as contact interactions between some homologous monomers, the so-called buttons (*Figure 2A*). For each pair of homologous chromosomes, positions along the genome were randomly selected as buttons with a probability ρ. Each 10-kbp bead *i* of chromosome *chr* is therefore characterized by a state $p_{chr,i}$ with $p_{chr,i} = 1$ if it is a button (= 0 otherwise) and $p_{chr,i} = p_{chr',i} = 1$ if *chr* and *chr'* are homologous. In addition, the first 1000 monomers of each chromosome were modeled as self-attracting centromeric and pericentromeric regions, the rest as neutral euchromatic regions. The energy of a given configuration was given by

$$H = \sum_{chr} k \sum_i \left(1 - cos\theta_{i,chr}\right) + \sum_{chr,chr',i,j} \left[E_p\delta_{chr,i;chr',j}\Delta_{chr,i;chr',j}p_{chr,i} + E_c\delta_{chr,i;chr',j}C_{chr,i}C_{chr',j}\right], \qquad \text{(S1)}$$

where *k* is the bending rigidity, $\theta_{i,chr}$ is the angle between the bond vectors *i* and *i+1* of chromosome *chr*, $\delta_{chr,i;chr',j} = 1$ if beads *i* from chromosome *chr* and *j* from *chr'* occupy nearest-neighbor sites (= 0 otherwise), $\Delta_{chr,i;chr',j} = 1$ if *i = j* and *chr* and *chr'* are homologous (= 0 otherwise), $C_{chr,i} = 1$ if bead *i* of *chr* is a (peri-)centromeric region, $E_p < 0$ is the contact energy between homologous buttons, $E_c < 0$ is the contact energy between centromeric beads.

The dynamics of the chains followed a simple kinetic Monte-Carlo scheme with local moves using a Metropolis criterion applied to *H*. The values of *k* (=1.5*kT*), *b* (=105 nm), $E_c$(=−0.1*kT*), $L_x = L_y$ (=2 μm), and $L_z$ (=4 μm) were fixed using the coarse-graining and time-mapping strategies developed in *Ghosh and Jost, 2018* for a 10 nm fiber model and a volumic density = 0.009 bp/nm$^3$ typical of *Drosophila* nuclei. For every set of remaining parameters (the button density ρ and the strength of pairing interaction $E_p$), 250 independent trajectories were simulated starting from compact, knot-free,

Rabl-like initial configurations (*Dernburg et al., 1996*): all centromeric regions were localized at random positions at the bottom of the simulation box, the rest of the chain being confined into a cylinder of diameter ~600 nm and height ~2 μm pointing toward the top of the box (see examples in *Figure 2C* and *Figure 2—video 1* and *2*). The distance between the centers of mass of homologous chromosomes, noted as $d_i$, typically varied between 0.5 μm and 3 μm. Each trajectory represented ~10 h of real time. To model the developmental pairing dynamics, we ran simulations in which $d_i$ was sampled from the distribution inferred from chromosome painting experiments (*Figure 2—figure supplement 2*).

To constrain model parameters, we compared the measured pairing dynamics (*Figure 4B*) to the model prediction. Specifically, for each parameter set, we computed a chi²-score between the predicted dynamics and experimental time points

$$chi^2 = (1/5) \sum_{t=3h}^{5.5h} \left[ \left(P_{pred}(t) - P_{exp,38F}(t)\right)^2 / \left(2\sigma_{exp,38F}^2(t)\right) + \left(P_{pred}(t) - P_{exp,53F}(t)\right)^2 / \left(2\sigma_{exp,53F}^2(t)\right) \right], \quad \text{(S2)}$$

with $P_{pred}(t)$ the predicted dynamics at developmental time $t$, $P_{exp,38F}(t)$ and $P_{exp,53F}(t)$ the experimental average dynamics for loci 38F and 53F at time $t$, respectively, and $\sigma_{exp,38F}(t)$ and $\sigma_{exp,53F}(t)$ their corresponding standard deviations at time $t$.

## DNA constructs and fly lines

Flies expressing a nuclear MCP-NLS-mCherry under the control of the nanos promoter were previously described (*Bothma et al., 2018*). To create flies expressing PCP-NoNLS-GFP, the plasmid pCASPER4-pNOS-eGFP-PCP-αTub3′UTR was constructed by replacing the MCP coding region of pCASPER4-pNOS-NoNLS-eGFP-MCP-αTub3′UTR (*Garcia et al., 2013*) with the coding region of PCP (*Larson et al., 2011*). Transgenic lines were established via standard P-element transgenesis (*Spradling and Rubin, 1982*). To create flies expressing MS2 or PP7 loops under the control of UAS, we started from plasmids piB-hbP2-P2P-lacZ-MS2-24x-αTub3′UTR (*Garcia et al., 2013*) and piB-hbP2-P2P-lacZ-PP7-24x-αTub3′UTR, the latter of which was created by replacing the MS2 sequence of the former with the PP7 stem loop sequence (*Larson et al., 2011*). The *hunchback* P2P promoter was removed from these plasmids and replaced by 10 copies of the UAS upstream activator sequences (*Brand and Perrimon, 1993*) and the *Drosophila* Synthetic Core Promoter (DSCP) (*Pfeiffer et al., 2010*). Recombinase-mediated cassette exchange (*Bateman et al., 2006*) was then used to place each construct at two landing sites in polytene positions 38F and 53F (*Bateman and Wu, 2008*; *Bateman et al., 2012a*). Flies carrying the GAL4 driver *nullo-GAL4,* which drives expression in all somatic cells during the cellular blastoderm stage of cell cycle 14, were a gift from Jason Palladino and Barbara Mellone. Flies carrying the GAL4 driver *R38A04-GAL4,* which drives expression in epidermal cells in germband-extended embryos (*Jenett et al., 2012*), were acquired from the Bloomington *Drosophila* Stock Center. Finally, the interlaced MS2 and PP7 loops under the control of the *hunchback* P2 enhancer and promoter (P2P-MS2/PP7-lacZ) were based on a previously described sequence (*Wu et al., 2014*).

To create embryos for analysis of pairing, mothers of genotype *10XUAS-DSCP-MS2; MCP-mCherry-NLS, PCP-GFP* were crossed to males of genotype *nullo-GAL4, 10XUAS-DSCP-PP7*. The resulting embryos are loaded with MCP-mCherry-NLS and PCP-GFP proteins due to maternal expression via the nanos promoter, and zygotic expression of *nullo-GAL4* drives transcription of MS2 and PP7 loops in all somatic cells starting approximately 30 min into cell cycle 14 (cellular blastoderm). For pairing analysis, both MS2 and PP7 transgenes were in the same genomic location, either position 38F or 53F, whereas for the negative control, MS2 loops were located at 38F and PP7 loops were located at 53F. To visualize pairing at later times in development, the mothers indicated above were instead crossed to males of genotype *10XUAS-DSCP-PP7; R38A04-GAL4*, where both MS2 and PP7 loops were located at position 38F. Finally, to visualize MS2 and PP7 loops derived from the same genomic location, mothers of genotype *MCP-mCherry-NLS, PCP-GFP* were crossed to P2P-MS2/PP7-lacZ located at position 38F.

## Embryo preparation and image acquisition

Embryos were collected at 25℃ on apple juice plates and prepared for imaging as previously described (*Garcia et al., 2013*). Mounted embryos were imaged using a Leica SP8 confocal

microscope, with fluorescence from mCherry and eGFP collected sequentially to minimize channel crosstalk. For each movie, the imaging window was 54.3 × 54.3 μm at a resolution of 768 × 768 pixels, with slices in each z-series separated by 0.4 μm. Z-stacks were collected through either 10 or 12 μm in the z plane (26 or 31 images per stack), resulting in a time resolution of approximately 27 or 31 s per stack using a scanning speed of 800 Hz and a bidirectional scan head with no averaging. For the pairing data in *Figure 3*, the imaging window was centered laterally for embryos that were pre-gastrulation; for post-gastrulation embryos, the imaging window centered on a dorsal view of the embryonic head region covering mitotic domains 18 and 20 (*Foe, 1989*), which shows minimal movements during gastrulation and germ band extension relative to other regions of the embryo. We compared pairing levels in these cells at 6 hr of development to that of cells in a posterior abdominal segment at the same time point and found them to be nearly identical (75.0% paired, n = 16 for anterior cells vs. 73.7%, n = 19 in posterior cells according to the definition of pairing in *Figure 3E*), confirming that cells from different regions of the embryo are roughly equivalent for pairing dynamics at this stage. For positive-control embryos with interlaced PP7 and MS2 loops driven by the *hunchback* promoter, embryos were imaged during cell cycle 13 and early cell cycle 14, and the imaging window was positioned laterally as previously described (*Garcia et al., 2013*). To assess pairing in late-stage embryos using the *R38A04-GAL4* driver, embryos were aged to approximately 11–12 hr and the imaging window was positioned laterally over an abdominal segment. For the developmental time course movies in *Figure 4*, imaging centered on mitotic domains 18 and 20 when these cells were in interphase. During time points when these domains were undergoing mitosis, an adjacent mitotic domain in interphase was imaged.

## Image analysis

All images were first run through the ImageJ plug-in Trainable Weka Segmentation (*Arganda-Carreras et al., 2017*) and filtered with custom classifiers to generate two separate channels of 3D segmented images that isolated fluorescent spots. These segmented spots were then fitted to a Gaussian with a nonlinear least squares regression to find the 2D center. Image z-stacks were then searched for any spots tracked for three or more contiguous z-slices and the r est were discarded. Additional manual curation was employed to confirm the accuracy of segmented images and to add any spots that were missed. An initial estimate of the center of each spot was set based on the z-slice in which the spot had the greatest maximum intensity within a predefined radius from its 2D center. These initial estimates were then used to seed a 3D Gaussian fit for each spot, the center of which was used for all distance calculation. This granted us not only sub-pixel resolution in x-y but also sub-z-slice resolution, allowing for more precision in the z coordinate, which would otherwise be limited by the 0.4 μm spacing between consecutive stacked images created by confocal imaging.

Raw image z-stacks for each time frame were also maximum projected in the channel containing nuclearly localized MCP-mCherry to create 2D maps of all the nuclei in frame. These nuclear projections were then segmented and tracked in Matlab, followed by manual curation to ensure that each nucleus was consistently followed. One tracked particle lineage from each channel was then assigned a distinct nucleus based on its proximity to that nucleus in the 2D map and the particles in each channel were considered homologous chromosomes of one another. Since absolute coordinates of assigned particles were not possible to obtain due to cellular rotation and motion, all distance calculations were done with the relative coordinates of each locus from its homolog; any cellular rotation or motion was assumed to be conserved between loci in the same cell.

For the data in *Figure 3*, we qualitatively scored each nucleus based on the measured distances between red and green signals over the time that the signals were observed: 'paired' nuclei showed small distances and little variation over time and 'unpaired' nuclei showed larger distances and greater variation over time. Nuclei that showed a transition from large distances and variation at earlier time points to smaller distances and variation at later time points were scored as 'pairing' traces and were not included in *Figure 3* (see *Figure 5*). In assessing the stability of the paired state, we included both 'paired' (n = 25) and 'pairing' (n = 13) nuclei from three embryos in the total number of nuclei (n = 38) assessed. In this analysis, we conservatively only included the observation time of 'paired' nuclei (> 8 hr of observation with no transition back to the unpaired state), although 'pairing' nuclei also remained in the paired state throughout the remaining observation time once they became paired.

In some traces, signal is temporarily lost, which could be due to either a loss of fluorescence of the MS2 or PP7 reporters caused by transcriptional bursting or due to one or both loci moving out of our imaging window. For paired loci, we randomly sampled six traces and found that only two had any missing frames, with the missing events due solely to loss of transcription (three lost frames out of 339 total tracked frames in the sampling). Therefore, missing frames do not significantly impact our measurements of paired loci. In the case of unpaired loci, where the relative movement of homologous loci is less restricted, there is a greater risk of systematically underestimating the mean distance between signals if missing frames are caused by at least one homolog moving out of the field of view. To investigate this, we randomly sampled six unpaired loci traces and found that four of the six traces had missing frames with at least one signal outside of the imaging window (54 outside-of-window frames out of 587 total tracked frames in the sampling). To probe the possible impact on the mean distance of these traces, we assumed that the distance between homologs in all the missing frames of the six sampled traces was 5 µm, corresponding to the average nucleus diameter. While it is unlikely that all our missing frames contained loci that were 5 µm apart, this approach gives us an upper bound of the possible impact of missing frames. We found a rather modest effect with an increase of the mean distance of ~12.5% (from 2.4 µm to 2.7 µm) that is unlikely to alter any of our conclusions.

To align the traces presented in *Figure 5* based on a time point when the loci become paired, we manually aligned all traces that had been qualitatively assessed as 'pairing' traces according to several values of threshold distance and consecutive frames below that threshold. We then optimized this exploration for values that provided qualitatively good alignment of traces but that excluded as few traces as possible in order to maximize the data available for analysis. The same criteria were applied to identify and align pairing traces from simulations.

All image analysis was done using custom scripts in Matlab 2019b unless otherwise stated. These scripts can be found at https://github.com/GarciaLab/mRNADynamics/ (*Garcia Lab, 2021a*, copy archived at swh:1:rev:a1b5c591656cae816ed6fc4a4e447c3bd375959c, *Garcia Lab, 2021b*).

## Chromosome painting

Embryos of genotype $w^{1118}$ were aged to 2–3 hr after embryo deposition, fixed, and subjected to DNA-FISH using 400 pmol of Oligopaint probes (*Beliveau et al., 2012*) targeting 2L and 2R (200 pmol of each probe; *Rosin et al., 2018*) as previously described (*Bateman and Wu, 2008*). Hybridized embryos were mounted in Vectashield mounting medium with DAPI (Vector Laboratories), and three-dimensional images were collected using a Leica SP8 confocal microscope. To establish initial inter-homolog distances, an image from an embryo in early interphase 14 (as judged by nuclear elongation [*Fung et al., 1998*]) and with high signal-to-noise ratio was analyzed using the TANGO image analysis plug-in for ImageJ (*Ollion et al., 2013*; *Ollion et al., 2015*; *Belevich et al., 2016*). After segmentation and assignment of each painted territory to a parent nucleus, distances between territories were measured from centroid to centroid in 3D. Since homologous chromosomes are labeled with the same color, when territories produce a continuous region of fluorescence, a distance of zero was assigned. A total of 48 nuclei were analyzed for each of 2L and 2R.

## Acknowledgements

We thank Florian Jug for help with an earlier version of the nuclear tracking software. We also thank Francesco Ferrari, Gary Karpen, Abby Dernburg, Cédric Vaillant, and members of the Jost group for fruitful discussions. HGG was supported by the Burroughs Wellcome Fund Career Award at the Scientific Interface, the Sloan Research Foundation, the Human Frontiers Science Program, the Searle Scholars Program, the Shurl and Kay Curci Foundation, the Hellman Foundation, the NIH Director's New Innovator Award (DP2 OD024541-01), and an NSF CAREER Award (1652236). DJ acknowledges Agence Nationale pour la Recherche (ANR-18-CE12-0006-03, ANR-18-CE45-0022-01) and ITMO Cancer (Plan Cancer 2014–2019, Biologie des Systèmes n°BIO2015-08) for funding and CIMENT infrastructure (supported by the Rhone-Alpes region, Grant CPER07 13 CIRA) for computational resources. JRB was supported by grants from the National Institutes of Health (P20 GM0103423 and R15 GM132896-01) and an NSF CAREER Award (1349779).

## Additional information

### Funding

| Funder | Grant reference number | Author |
|---|---|---|
| Burroughs Wellcome Fund | Career Award at the Scientific Interface | Hernan G Garcia |
| Alfred P. Sloan Foundation | Sloan Research Fellowship | Hernan G Garcia |
| Human Frontier Science Program | | Hernan G Garcia |
| Searle Scholars Program | | Hernan G Garcia |
| Shurl and Kay Curci Foundation | | Hernan G Garcia |
| Hellman Foundation | | Hernan G Garcia |
| National Institutes of Health | DP2 OD024541-01 | Hernan G Garcia |
| National Science Foundation | 1652236 | Hernan G Garcia |
| Agence Nationale de la Recherche | ANR-18-CE12-0006-03 | Daniel Jost |
| Agence Nationale de la Recherche | ANR-18-CE45-0022-01 | Daniel Jost |
| ITMO University | BIO2015-08 | Daniel Jost |
| National Institutes of Health | P20 GM0103423 | Jack R Bateman |
| National Institutes of Health | R15 GM132896-01 | Jack R Bateman |
| National Science Foundation | 1349779 | Jack R Bateman |

The funders had no role in study design, data collection and interpretation, or the decision to submit the work for publication.

### Author contributions

Myron Barber Child VI, Data curation, Software, Formal analysis, Investigation, Visualization, Writing - original draft; Jack R Bateman, Conceptualization, Data curation, Formal analysis, Supervision, Investigation, Visualization, Methodology, Writing - original draft, Project administration; Amir Jahangiri, Software, Methodology; Armando Reimer, Nicholas C Lammers, Software; Nica Sabouni, Diego Villamarin, Grace C McKenzie-Smith, Justine E Johnson, Investigation; Daniel Jost, Conceptualization, Resources, Software, Supervision, Funding acquisition, Validation, Investigation, Visualization, Methodology, Writing - original draft; Hernan G Garcia, Conceptualization, Resources, Supervision, Funding acquisition, Visualization, Methodology, Writing - original draft

### Author ORCIDs

Myron Barber Child VI (iD) https://orcid.org/0000-0001-8563-0842
Jack R Bateman (iD) https://orcid.org/0000-0002-8782-5958
Nicholas C Lammers (iD) http://orcid.org/0000-0001-6832-6152
Diego Villamarin (iD) http://orcid.org/0000-0003-3265-1740
Daniel Jost (iD) https://orcid.org/0000-0002-9877-6864
Hernan G Garcia (iD) https://orcid.org/0000-0002-5212-3649

### Decision letter and Author response

Decision letter https://doi.org/10.7554/eLife.64412.sa1
Author response https://doi.org/10.7554/eLife.64412.sa2

## Additional files

### Supplementary files
• Transparent reporting form

### Data availability

Modeling code is available at: https://github.com/physical-biology-of-chromatin/Homologous_pairing (copy archived at https://archive.softwareheritage.org/swh:1:rev:09c00ff8e63d6f-be812660771fd2d22df277aa1a). Custom Matlab 2019b a image analysis scripts can be found at: https://github.com/GarciaLab/mRNADynamics/ (copy archived at https://archive.softwareheritage.org/swh:1:rev:a1b5c591656cae816ed6fc4a4e447c3bd375959c). Raw figure files of relevant plots are available at: https://doi.org/10.5281/zenodo.5063001. Samples of generated data used in this study are included in the manuscript and in supporting files available at: https://doi.org/10.5061/dryad.3j9kd51j5.

The following datasets were generated:

| Author(s) | Year | Dataset title | Dataset URL | Database and Identifier |
|---|---|---|---|---|
| Child MB, Bateman JR, Jahangiri A, Reimer A, Lammers NC, Sabouni N, Villamarin D, McKenzie-Smith GC, Johnson JE, Jost D, Garcia HG | 2021 | Live imaging and biophysical modeling support a button-based mechanism of somatic homolog pairing in Drosophila | https://doi.org/10.5061/dryad.3j9kd51j5 | Dryad Digital Repository, 10.5061/dryad.3j9kd51j5 |
| Child MB, Bateman JR, Jahangiri A, Reimer A, Lammers NC, Sabouni N, Villamarin D, McKenzie-Smith GC, Johnson JE, Jost D, Garcia HG | 2021 | Live imaging and biophysical modeling support a button-based mechanism of somatic homolog pairing in Drosophila | https://doi.org/10.5281/zenodo.5063001 | Zenodo , 10.5281/zenodo.5063001 |

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
