## [Decision Letter]

**Acceptance summary:**

The way homologous chromosomes identify one another and become paired is an intriguing phenomenon that has a long history of study, yet the molecular mechanism remains unclear. Recent studies have led to a phenomenological button model for homolog pairing, which hypothesises that pairing is initiated at discrete sites along the length of each chromosome. The authors investigate this idea rigorously using biophysical modelling and live imaging. They constructed a simple polymer model with buttons distributed along the chain that possess locus-specific interactions, and thoroughly investigated its property via stochastic simulation in 3D. Their study confirms that homolog-specific interactions are necessary for homolog pairing. The authors went on to perform live imaging of pairing dynamics at two selected loci, using the fluorescent signal from nascent mRNA at the corresponding locus, and found satisfactory agreement with the model. Their study supports a button mechanism for homolog pairing, where stable pairing is initiated by reversible random encounters that are propagated chromosome-wide. This work suggests that active processes are not necessary to explain pairing and paves the way for further investigating the molecular mechanism of such a pairing phenomenon.

**Decision letter after peer review:**

Thank you for submitting your article "Live imaging and biophysical modeling support a button-based mechanism of somatic homolog pairing in *Drosophila*" for consideration by *eLife*. Your article has been reviewed by 3 peer reviewers, and the evaluation has been overseen by a Reviewing Editor and Naama Barkai as the Senior Editor. The following individual involved in review of your submission has agreed to reveal their identity: Shou-Wen Wang (Reviewer #2).

The referees unanimously found your work to be interesting. and highly relevant to the field of somatic homolog pairing in *Drosophila*. There is a number of points that need to be addressed before a final decision can be made regarding publication.

1. The theoretical model here described is a close variant of a model introduced some years ago in Genetics 179, 717 (2008). As its title clearly shows ("A Thermodynamic Switch for Chromosome Colocalization"), that paper envisaged a mechanism whereby the interaction energy between specific regions on the homologs thermodynamically stabilises their random encounters, producing a transition from an unpaired to a paired state. I think that the authors should clearly acknowledge that previous paper in their manuscript as required by the best practices of the scientific community.

2. While the authors convincingly show that the button model can explain homolog pairing, their data show areas of quantitative disagreement, which might highlight the need for future improvement of the modeling and experimental design. Specifically: the model does not accurately reproduce the observed pairing probability over developmental time (Figure 4B). The author already commented on the discrepancy at time=6h. I found the discrepancy at t=0h also puzzling: while the observed pairing probability is around 0 for both loci, the model predicts a 10% pairing probability at t=0. A comment or explanation here will be very useful for the readers.

3. Similarly, in Figure 5, while the model accurately reproduced the post pairing behavior under constrained parameters, the pre-pairing dynamics are not well reproduced: the observed inter-locus distance decreases linearly with time, while the predicted decrease has a rather nonlinear pattern, speeding up as the pairing is being established. An explanation here is useful.

4. The size of buttons should be addressed – small vs. large buttons. The authors build their model around a 10 kb button size. It is not clear why they only tested this button size. In the Rowley and Alhaj Abed studies, they conducted HiC which reflects stable state pairing, where the actions of multiple buttons could drive pairing. Based on their findings, they predict "small" buttons of insulator size (2-10kb). Viets and colleagues conducted functional transgene studies that identify the sufficiency of regions to drive pairing. Their studies predict "large" buttons of ~90 kb.

The assumption of this 10kb button size in this paper imply that the drivers of pairing are a number of small elements whose percentage determines affinity. However, this assumption does not take into account the counter model that buttons are larger ~90 kb elements. Considering that the Viets study is done by testing the pairing capacity of elements, the authors should consider this "large button" hypothesis in their model.

Along these lines, the authors conclude that pairing readily occurs at roughly 70% density (Figure 2D middle), suggesting 70 kb buttons that resemble the "large" 90 kb buttons. The authors should reconcile these data and test both models.

5. The spatial correlations between distant buttons should be discusses in more depth. The extent to which local versus distant effects of buttoning events are included in the model should be clarified the potential implications of distant effects should be discussed. Related to this, the zipping process, where a paired locus facilitates the pairing at neighboring loci, is a prediction unique to the button model. This cannot be tested directly by the current experimental design. Its test requires observing the pairing dynamics of multiple neighboring loci along the same chromosome. While this goes beyond the scope of this paper, it is worth mentioning this limitation in the paper.

6. Figure 3 is confusing and would benefit from information from Figure 5-sup 1. In figure 3, C and D are not presented in a manner that is ideal for the reader:

i. Why does the unpaired control end at ~25 minutes?

ii. The color codes for the graphs are confusing. Unpaired control and paired control are from two different experimental conditions and should be in two different colors.

iii. The data from Figure 5-sup1 should be included in Figure 3. Specifically, more individual traces to represent the data (from A and B) and the average traces for each condition (as in C, but for all conditions).

7. Why is signal lost? At some points, signal is lost in their MS2 and PP7 experiments. The authors should clearly state why this occurs and what it means for their analysis. Is it because transcription is bursty and these are breaks in transcription or is it out of the plane? If it is out of the planes of imaging, how does this affect the analysis, especially as these could potentially lead to greater distances between dots.

8. A full experimental confirmation of the model could be attained by perturbing the proposed mechanism. For instance, it could be shown that if the interaction energy between the buttons is reduced below a threshold value, pairing doesn't occur anymore. That could be experimentally achieved by interfering with the molecular elements associated to the interaction between the buttons, for example, by targeted nested deletions of those genomic regions or by titrating out the related pairing factors. A discussion of such experiments in the paper would be welcome, as they appear quite feasible and would provide a clear proof of the proposed mechanism.

---

## [Author Response]

The referees unanimously found your work to be interesting. and highly relevant to the field of somatic homolog pairing in *Drosophila*. There is a number of points that need to be addressed before a final decision can be made regarding publication.1. The theoretical model here described is a close variant of a model introduced some years ago in Genetics 179, 717 (2008). As its title clearly shows ("A Thermodynamic Switch for Chromosome Colocalization"), that paper envisaged a mechanism whereby the interaction energy between specific regions on the homologs thermodynamically stabilises their random encounters, producing a transition from an unpaired to a paired state. I think that the authors should clearly acknowledge that previous paper in their manuscript as required by the best practices of the scientific community.

We agree that we should have cited this paper by Nicodemi, Panning, and Prisco. We did cite a different paper by these authors published in the same year in a different journal that also describes their model (Results, lines 174-178), along with works by Penfold et al., 2012 and Marshall and Fung, 2016 that also explore polymer models in the context of meiotic pairing. We have added reference to the Nicodemi et al. *Genetics* paper as suggested, and added text to the Discussion (lines 554-556) and Results sections (lines 127-135) to more clearly acknowledge these works.

2. While the authors convincingly show that the button model can explain homolog pairing, their data show areas of quantitative disagreement, which might highlight the need for future improvement of the modeling and experimental design. Specifically: the model does not accurately reproduce the observed pairing probability over developmental time (Figure 4B). The author already commented on the discrepancy at time=6h. I found the discrepancy at t=0h also puzzling: while the observed pairing probability is around 0 for both loci, the model predicts a 10% pairing probability at t=0. A comment or explanation here will be very useful for the readers.

For the experimental data at position 53F (blue points in Figure 4B), we see considerable overlap in the estimated error between the simulations (light grey) and the observed values (blue whiskers) in the first time point, but we agree that the data for 38F show a value for the measured percentage of paired nuclei below the prediction. We provide two possibilities to account for this. First, it is possible that, in our simulations, we are overestimating the number of nuclei with low values of the initial distance *d_i_*. This overestimation would increase the number of nuclei that are paired by chance at the beginning of the simulation. The chromosome paint experiment that we base our estimations on has a resolution of roughly 1 µm, and we model *d_i_
*by fitting a Gaussian to those data (see Figure 4—figure supplement 2). However, it is possible that the true values of distances are not described by this Gaussian distribution at values below 1 µm. Secondly, as we mention in the Discussion, observations in the blastoderm stage by Lim et al. showed that pairing is generally unstable at that stage. Therefore the lower-than-expected values at our first time point, which is approximately mid- to late-stage blastoderm, may be a consequence of this transition from early/unstable to late/stable pairing that is not accounted for by our model. We have included a discussion of this caveat in the Results section (lines 359-363).

3. Similarly, in Figure 5, while the model accurately reproduced the post pairing behavior under constrained parameters, the pre-pairing dynamics are not well reproduced: the observed inter-locus distance decreases linearly with time, while the predicted decrease has a rather nonlinear pattern, speeding up as the pairing is being established. An explanation here is useful.

We agree that the observed *median* pre-pairing dynamics (black full lines in Figure 5C-F) appears more linear than the corresponding predictions of the model (colored full lines in Figure 5F). However, we would like to note that the predicted and experimental trajectories are highly stochastic as quantified by the large error bars in Figures 5C-F. Within this variability, predictions are fully compatible with the experimental data. Moreover, the experimental curve was computed using only 14 trajectories due to the difficulty of capturing these events, and was smoothed (Figure 5 figure supplement 1C) in order to capture the main behavior of the dynamics while filtering for large fluctuations inherent to very low statistics. In contrast, the predictions are based on more than 10,000 events and are statistically better defined. In a new figure panel (Figure 5—figure supplement 2C), we tested whether having low statistics and smoothing can slightly perturb the overall shape of the median pre-pairing dynamics of the predictions. In particular, we provided a representative example (where we randomly pick 14 simulated trajectories and apply the same treatment as for experimental data) that exhibits a ‘linear’ pre-pairing dynamics as observed in experiments. We have added references in the main text to Figure 5—figure supplement 2C (lines 396-398, 440, and 990-991) where we now clearly discuss this point.

4. The size of buttons should be addressed – small vs. large buttons. The authors build their model around a 10 kb button size. It is not clear why they only tested this button size. In the Rowley and Alhaj Abed studies, they conducted HiC which reflects stable state pairing, where the actions of multiple buttons could drive pairing. Based on their findings, they predict "small" buttons of insulator size (2-10kb). Viets and colleagues conducted functional transgene studies that identify the sufficiency of regions to drive pairing. Their studies predict "large" buttons of ~90 kb.The assumption of this 10kb button size in this paper imply that the drivers of pairing are a number of small elements whose percentage determines affinity. However, this assumption does not take into account the counter model that buttons are larger ~90 kb elements. Considering that the Viets study is done by testing the pairing capacity of elements, the authors should consider this "large button" hypothesis in their model.Along these lines, the authors conclude that pairing readily occurs at roughly 70% density (Figure 2D middle), suggesting 70 kb buttons that resemble the "large" 90 kb buttons. The authors should reconcile these data and test both models.

While it is certainly a question of great interest to all of us, we hope that the reviewer will agree that dissecting the exact molecular nature of buttons and their positions along the genome is beyond the scope of the paper. In the Discussion section of the first version of the manuscript, we discussed some relevant hypotheses. In particular, in line with recent analyses showing that ‘buttons’ are enriched in architectural proteins (Rowley et al. 2019) or insulators (Viets et al. 2019), we tested the possibility that buttons containing multiple binding sites for architectural proteins or insulators could support pairing (Figure 2—figure supplement 4A-C).

We thank the reviewers for raising the issue of the ‘button size’. Indeed, in their work, Viets et al. suggest that pairing units (chromosomal segments supporting pairing) may need to be of a certain size (TAD size) to allow efficient pairing, based on the monitoring of pairing between transgenes on heterologous chromosomes. Actually, we do not believe that this suggestion of large pairing units by Viets et al. contradicts the Abed/Rowley observations that buttons (elementary loci actually driving pairing) are ‘small’. Indeed, our model suggests that for pairing between two regions to ensue, enough ‘small’ buttons inside these regions may be needed (for a given strength of interaction per button). The TAD-sized transgenes that were capable of inducing ectopic pairing in the Viets et al. study may actually correspond to pairing units containing several ‘small’ buttons. This may explain why only ‘large’ transgenes that carry enough small buttons can be paired.

At the end of their paper (Figure 7 I-K), Viets et al. proposed three possibilities regarding the position and nature of the pairing units (or ‘large’ buttons): (1) they may correspond to TAD boundaries. This hypothesis would be equivalent to our specific button model at low button density where small buttons would correspond to TAD boundaries (about 300 TAD boundaries per chromosome, i.e. a button density of ~15 % in our terminology). We showed that such low button density is also compatible with pairing if the strength of interaction is strong enough (Figure 4C of our manuscript). (2) Pairing units may be made by a unique combination of insulators (“insulator code”). This hypothesis exactly corresponds to the combinatorial model investigated in Figure 2—figure supplement 4A-C where we showed that this is indeed compatible with pairing. (3) Pairing units may represent unique microcompartments. To test this possibility, we launched a new series of simulations where consecutive ‘small’ buttons along the genome may form specific microcompartments that would correspond to ‘large’ buttons (Figure 2—figure supplement 4D-E). We showed that this possibility is also compatible with pairing as long as the pairing units are not too large (size below 750kbp), which is the case if pairing units overlap with TADs.

In addition to the Supplementary panels (Figure 2—figure supplement 4D-E), we have added text in the Discussion section to present these points (lines 541-544).

5. The spatial correlations between distant buttons should be discusses in more depth. The extent to which local versus distant effects of buttoning events are included in the model should be clarified the potential implications of distant effects should be discussed. Related to this, the zipping process, where a paired locus facilitates the pairing at neighboring loci, is a prediction unique to the button model. This cannot be tested directly by the current experimental design. Its test requires observing the pairing dynamics of multiple neighboring loci along the same chromosome. While this goes beyond the scope of this paper, it is worth mentioning this limitation in the paper.

We agree with the reviewers that, beyond the description of the zippering process, we did not quantify or discuss spatial correlations in our previous manuscript. To correct for that, we first computed the cross-correlation of the pairing status of two loci separated by a given genomic distance (new Figure 2—figure supplement 3A). We observed that the pairing of loci separated by a large genomic distance (Mbp) remains correlated and that this correlation grows as the global pairing of the chromosome increases. We also launched a new series of simulations where we removed all the buttons present in a given chromosomal segment (new Figure 2—figure supplement 3B). Compared to the ‘wild-type’ situation (no removal), we observed that the local deletion of buttons impacts the pairing probability at long distances—up to 1 Mbp from the region where buttons were removed. These new analyses suggest that the pairing of homologous loci involved large-scale effects, which we discuss now in detail in the new version of the manuscript in the Discussion section (lines 465-468). As suggested by the reviewers, we also add text in the Discussion section (lines 476-480) to mention the limitation of the current experimental approach and the perspective of experiments to simultaneously monitor several pairs of loci to challenge the ziperring process predicted by the button model.

6. Figure 3 is confusing and would benefit from information from Figure 5-sup 1. In figure 3, C and D are not presented in a manner that is ideal for the reader:i. Why does the unpaired control end at ~25 minutes?

The data for Figure 3 was generated from videos of different durations, varying from approximately 30 to 60 minutes. Furthermore, since nuclei are moving relative to the field of view as the embryo develops, each nucleus may be tracked for less time than the entire length of the video. In the case of panel 3C, the unpaired control was generated from a 30-minute video, whereas the trace showing interhomolog distances for unpaired homologs was generated from a longer video. In an effort to make this figure less confusing, we have shortened the representative trace from unpaired homologs such that both traces in Figure 3C (and in Figure 3—figure supplement 2A-B) are the same length, and have clarified in the legend that tracking time for each nucleus varies depending on the length of the video and the time that the nucleus is in the frame of view.

ii. The color codes for the graphs are confusing. Unpaired control and paired control are from two different experimental conditions and should be in two different colors.

The colors have been made consistent throughout the figure to increase clarity.

iii. The data from Figure 5-sup1 should be included in Figure 3. Specifically, more individual traces to represent the data (from A and B) and the average traces for each condition (as in C, but for all conditions).

A new Figure 3—figure supplement 1 now shows a representative sample of individual traces for the experiment featured in Figure 3, along with median and interquartile ranges for each condition.

7. Why is signal lost? At some points, signal is lost in their MS2 and PP7 experiments. The authors should clearly state why this occurs and what it means for their analysis. Is it because transcription is bursty and these are breaks in transcription or is it out of the plane? If it is out of the planes of imaging, how does this affect the analysis, especially as these could potentially lead to greater distances between dots.

In some traces, signal is indeed temporarily lost. As the reviewer suggests, this could be due to either a loss of fluorescence of the MS2 or PP7 reporters caused by transcriptional bursting or the loci moving out of our imaging window. For paired loci, we randomly sampled six traces and found that only two had any missing frames, with the missing events due solely to loss of transcription (3 lost frames out of 339 total tracked frames in the sampling). Therefore, missing frames do not significantly impact our measurements of paired loci. In the case of unpaired loci, where the relative movement of homologous loci is less restricted, there is a greater risk of systematically underestimating the mean distance between signals if missing frames are caused by at least one homolog moving out of the field of view. To investigate this, we randomly sampled six unpaired loci traces, and found that four out of the six traces had missing frames with at least one signal outside of the imaging window (54 outside-of-window frames out of 587 total tracked frames in the sampling). To probe the possible impact on the mean distance of these traces, we assumed that the distance between homologs in all the missing frames of the six sampled traces was 5 µm, corresponding to the average nucleus diameter. While it is unlikely that all our missing frames contained loci that were 5 µm apart, this approach gives us an upper bound of the possible impact of missing frames. We found a rather modest effect with an increase of the mean distance of ~12.5% (from 2.4 µm to 2.7 µm) that is unlikely to alter any of our conclusions.

We therefore acknowledge the limitations of our imaging system and the potential for systematic underestimation in unpaired traces, but find no evidence of impact on our measurements of paired traces. Thus, it is unlikely that underestimation due to missing frames would alter any of our conclusions. We have now made this point clear in the Image Analysis section of the Material and Methods (lines 738-754).

8. A full experimental confirmation of the model could be attained by perturbing the proposed mechanism. For instance, it could be shown that if the interaction energy between the buttons is reduced below a threshold value, pairing doesn't occur anymore. That could be experimentally achieved by interfering with the molecular elements associated to the interaction between the buttons, for example, by targeted nested deletions of those genomic regions or by titrating out the related pairing factors. A discussion of such experiments in the paper would be welcome, as they appear quite feasible and would provide a clear proof of the proposed mechanism.

In the previous version of our manuscript, we briefly described possible future directions (last paragraph of the Discussion). We agree with the reviewers that perspective experiments challenging the model predictions could be better discussed. We have completed the text at the end of the Discussion section (lines 571-576), discussing possible experiments to test the model (in addition to the simultaneous imaging of several homologous pairs of loci, see Point 5). In particular, we discussed the possibility of mutating or titrating candidate insulators or architectural proteins involved in pairing to challenge the role of the strength of interactions on pairing, as well as the possibility to do targeted deletion of buttons to address the question of button density and to investigate distal effect of local perturbations (see also Point 5 of this rebuttal above and the new Figure 2—figure supplement 3B where we investigate by simulations the consequences of targeted deletions on pairing).